

# Functional and health promoting inherent attributes of *Enterococcus hirae* F2 as a novel probiotic isolated from the digestive tract of the freshwater fish *Catla catla*

Mohd Adnan[1], Mitesh Patel[2] and Sibte Hadi[3]

[1] Department of Clinical Nutrition, College of Applied Medical Sciences, University of Hail, Hail, Saudi Arabia
[2] Bapalal Vaidhya Botanical Research Centre, Department of Biosciences, Veer Narmad South Gujarat University, Surat, Gujarat, India
[3] School of Forensic and Applied Sciences, University of Central Lancashire, Preston, Lancashire, United Kingdom

Corresponding author
Mohd Adnan,
drmohdadnan@gmail.com,
mo.adnan@uoh.edu.sa

## ABSTRACT

**Background**. Probiotic microorganisms are gaining global importance because of their use in the preparation of a nutraceutical or in the treatment of infections. As per the health industry demand, there is an urgent need for exploring new indigenous probiotic strains with its specific origin due to variation in gut microflora, different food habits and specific host-microbial interactions. The main objective of the present study was to isolate and identify a novel probiotic *Enterococcus* strain from the gut of *Catla catla* fish and evaluate its potentiality as a potent probiotic.

**Methods**. The whole study was designed with the isolation of novel lactic acid bacterial strain from the gut of *Catla catla* fish with their biochemical and molecular identifications. The potentiality of the isolated strain as a potent probiotic was carried out according to the parameters described in FAD/WHO guidelines for the evaluation of probiotics in food.

**Results**. The isolated strain was confirmed as *Enterococcus hirae* F2 on the basis of various biochemical and 16s rRNA gene sequencing methods. *Enterococcus hirae* F2 was able to survive under highly acidic and bile salt concentration with the ability for the production of lipase and Bsh enzyme. It was also able to survive under simulated gastrointestinal conditions with the inhibition ability of various pathogens. The antioxidant potentiality with the cell surface hydrophobicity and cell aggregation ability confirms its potentiality as a potent probiotic. All the results detail the potency of *Enterococcus hirae* F2 as a novel probiotic for a safer use.

**Discussion**. The isolation of *Enterococcus hirae* with probiotic potential from the gut of fish is a new approach and done for the first time. However, the whole study concluded that the isolated strain might be used as a novel probiotic in the food industry for the production of new probiotic products which imparts health benefits to the host.

## INTRODUCTION

Today, consumers are aware about the connection among way of life, well-being and healthy diet. This explains the rising demand for products that can upgrade health beyond basic nutrition. As an outcome of this, expanded exploration is going on in the development of new nutritional supplementation methodologies in which different health and growth promoting compounds such as prebiotics, probiotics, phytobiotics, synbiotics and other utilitarian dietary supplements have been assessed (*Denev, 2008*). Probiotics are live microorganisms that, when administered in adequate amounts, promote health advantages to the host, especially to the digestive system (*FAO/WHO, 2002*). Probiotic bacteria are vital and provide beneficial nutritional impact on host's health for healthy gastrointestinal functions. The most essential properties of a probiotic encompasses flexibility and resistance to extreme acidic and bile salt conditions present in the stomach and small intestine (*Pereira, McCartney & Gibson, 2003*), high adhesion properties towards epithelial cells (*Kravtsov et al., 2008*; *Espeche et al., 2012*) and the ability to hinder the pathogen attachment by particular rivalry (*Reid, 2006*).

Throughout the years, many strains of bacteria have been found with probiotic properties, primarily consisting of lactic acid engendering *bacteria i.e., Lactococci, Enterococci, Streptococci, Bifidobacteria, Lactobacilli.* They predominantly harbour gut of animals, air, water, food, soil etc. Due to the valuable products synthesized by them with special inherent antimicrobial, antioxidant properties, they are now found in wide application. Since 2005, in United States, West Europe, Japan and India the consumption of probiotic based functional dairy products has increased by 12%, and in between 2014 and 2016 more than 90 innovative yogurt products launched worldwide (*Zenith, 2016*).

Probiotics confers a good number of health benefits such as modulation of immune system, resistance to infectious diseases (*Nomoto, 2005*), malignancy against colon and bladder tumours, counteractive action of osteoporosis, prevention of urogenital diseases, decrease of hypercholesterolemia, alleviation of constipation, protection against traveller's diarrhoea, and so on (*Lourens & Viljoen, 2001*).

Probiotics is a new approach to fight against various pathogenic bacteria. *B. lactis* HN019 was found to lessen the severity of diseases caused by *Salmonella* and *Escherichia coli* O157: H7 with reducing the severity of diarrhoea in human infants (*Shu & Gill, 2001*). Probiotic bacteria prevent the attachment of pathogenic bacteria with the epithelial cells by competing with them and reduced the inflammatory bowel diseases (*Hooper et al., 2001*). Infants with diarrhoea were treated with a formula containing *L. Reuteri* 55730 and *Bifidobacterium lactis* (*Weizman, Asli & Alsheikh, 2005*). Anti-carcinogenic and anti-mutagenic activity has also been reported by some probiotic strains due to the induction of pro-inflammatory and secretory responses and their ability to inactivate and metabolize mutagenic compounds (*Geier, Butler & Howarth, 2006*). Recent studies also indicated the beneficial effect of probiotics against hypertension with lowering the blood cholesterol level via converting milk protein into bioactive peptides (*Goel et al., 2006*; *Xu et al., 2008*).

*Enterococci* are lactic acid bacteria, which comprise more than 30 species and extensively dispersed in nature (*Eaton & Gasson, 2001*). Some *Enterococci* species can repress a few

pathogens as a result of the presence of bacteriocin and they are also part of the complex intestinal microflora of fish (*Giraffa, 2002*; *Chingwaru, Mpuchane & Gashe, 2003*; *Valenzuela et al., 2010*; *Hagi et al., 2004*). Lactic acid bacteria isolated from fish generally confer a selective advantage of tolerance to high salt concentration because of the salting process of fish during production (*Harwood, Whitlock & Withington, 2000*). The genus of *Enterococcus* assumes a recognized part in the improvement of organoleptic attributes in fermented foods. In the food industry, *Enterococci* have important implications and share various other valuable biotechnological qualities (*Jensen et al., 1975*; *Leroi, Foulquie-Moreno & De Vuyst, 2003*).

Many functional probiotic lactic acid bacteria are isolated from the fish and, when ingested in sufficient quantities, impart favourable host health (*Ringo & Olsen, 1999*). Indian major carp (*Catla catla*) is most popular and economic important fish in South Asia. Not much information related to the microbiota of this fish is known. In past few years, large number of probiotic strain were isolated and studied from various sources (*Hairul Islam et al., 2011*; *Vidhyasagar & Jeevaratnam, 2013*). However, a thorough review of literature revealed that only few studies have been carried out in obtaining the probiotics from the gut of *Catla catla* with very less experimental evidence. The primary objective of this study was the isolation of a new potent probiotic strain from the gut of *Catla catla*. Isolation of a specific strain of *Enterococcus* as a novel probiotic can be one of the first efforts in developing Indian major carp culture, which will be used as a new probiotic feed.

## MATERIALS AND METHODS

### Collection and extraction of fish gut

Freshwater fish *Catla catla* (1.5–2 kg) were collected from the local fish market. They were tightly packed in separate pre-sterile polyethylene bags following the aseptic collection of samples. The samples were then placed in iceboxes and were immediately transported to the laboratory where fishes were first washed with sterile distilled water and then surface sterilization by using 70% alcohol. The guts of the fish were extracted under aseptic conditions and washed with 0.85% normal saline. After extraction, gut samples were homogenized in normal saline, followed by enrichment in de Man, Rogosa and Sharpe broth (MRS) (Hi-Media®, India) and incubated anaerobically at 30 °C for 24 h (*Rengpipat, Rueangruklikhit & Piyatiratitivorakul, 2008*).

### Isolation and screening of potential probiotic lactic acid bacteria from fish gut

Lactic acid bacteria were isolated using spread plate technique from the gut samples of the fishes, which were enriched overnight in MRS broth previously. 0.1 ml aliquot of $10^{-5}$, $10^{-6}$, $10^{-7}$ diluted samples were spread onto MRS agar plates with bromocresol purple (BCP) and incubated anaerobically at 30 °C for 48 h. At the end of incubation, colonies with yellow colour were selected for morphological examination under microscope, which is a characteristic of lactic acid bacteria on MRS agar plates with BCP. Lactic acid bacteria were first screened for Gram's reaction and catalase test. Gram positive and catalase negative

isolates were selected for further study. These isolates were preserved in 20% glycerol (Sigma-Aldrich) at −20 °C.

## Molecular screening and identification of *E. hirae* F2 strain

Identification of *E. hirae F2* was carried out by both conventional biochemical tests as well as by 16S ribosomal RNA sequencing. The culture F2 was identified up to genus level according to the known biochemical methods protocol (*Harrigan & Margaret, 1970*; *Sneath et al., 1986*; *Stiles & Holzapfel, 1997*). This includes Gram's reaction, test for catalase enzyme, differentiation in homo/heterofermentative, production of ammonia from arginine, growth at different NaCl concentrations and different temperatures. However, F2 culture was also checked for its various sugar fermentation profile.

### Identification by 16s rRNA Gene Sequencing
#### Extraction and quantification of DNA

Genomic DNA of *E. hirae* F2 was extracted, using bacterial genomic DNA kit (GenElute™, Sigma-Aldrich, Bangalore, India) and quantification was done according to the described method (*Sambrook, Fritch & Maniatis, 1982*). 10 µL of extracted DNA was dissolved in 30 µl of Tris buffer (pH-8.0) and OD was taken at 260 and 280 nm (UV-1800 Shimadzu Spectrophotometer; Shmadzu, Kyoto, Japan). OD of the samples showed between 1.6-1.8, which affirms the purity and integrity of the samples. However, purity of extracted genomic DNA was also checked by agarose gel electrophoresis (0.8%).

#### PCR amplification of 16S rRNA gene

16S rRNA region of *E. hirae* F2 was amplified by using 1X (final concentration) ReadyMix™ Taq PCR Reaction Mix (Sigma-Aldrich, Bangalore, India) and template DNA (50 ng/ µL). The reaction was carried out in Thermal cycler (Applied Biosystems Veriti; Applied Biosystems, Foster City, CA, USA). Pair of universal primers 27f and 1492r was used to amplify 16S rRNA region. 27f 5′AGAGTTTGATCMT GGCTCAG3′ used as a forward and 1492r 5′CGGTTACCTTGTTACGACTT3′ was used as a reverse primer (*Weisburgv et al., 1991*). PCR reaction mixture contained 1 x reaction mixture (10 µL), forward primer (1 µL), reverse primer (1 µL), genomic DNA template (2 µL), nuclease free water (6 µL). PCR program was adjusted as: Initialization at 95 °C for 5 min, 30 cycles of denaturation at 95 °C for 1 min, annealing at 55 °C for 1 min, extension at 72 °C for 1 min; and a final elongation step at 72 °C for 3 min followed by hold at −4 °C for ∞ time. Amplified PCR products were detected on agarose gel (1%) by electrophoresis staining with ethidium bromide and visualizing under UV light. The gels were analyzed by using the software Image lab version 3.0 (Bio Rad, Hercules, CA, USA). Purification of amplified PCR product was done using GenElute™ PCR Clean-up kit (Sigma®, India).

#### Cycle sequencing, analysis and sequence submission

Sequencing was carried out using Big Dye® Terminator v 3.1 Cycle sequencing kit with 27f and 1492r primers according to manufacturers protocol. Nucleotide sequencing was carried out in house with an ABI Prism 3130 automatic sequencer (Applied Biosystems, USA). Sequence analysis was done using sequencing analysis software version 5.4 (Applied Biosystems, Foster City, CA, USA) and BioEdit 7.2.5. These sequences were subjected to

sequence match analysis using Basic Local Alignment Search Tool (BLAST) on NCBI. The sequences were submitted to the NCBI GenBank datase under GenBank ID KF496213.1.

## Probiotic potential and assessment of *E. hirae* F2

### Bile salt tolerance

Ability of *E. hirae* F2 to grow in the presence of bile salt was determined by both tube as well as plate method containing different amount of bile salts. In the tube method, fresh culture (1%) was inoculated in to MRS broth containing different concentration of bile salts (Sigma-Aldrich, Bangalore, India) from 0.2 to 2%. Tubes were incubated anaerobically at 37 °C for 24 h, growth was confirmed by taking OD of the tubes at 560 nm after 24 h and then compared with a control tube (without bile salts) (*Vinderola & Reinheimer, 2003*). However, the ability of F2 culture to survive in bile salt was also assessed in terms of viable count by plating culture on MRS agar plates by withdrawing aliquots at 0, 30, 60, 90, 120 min time interval and incubated at 37 °C for 24 h. Counts of viable colonies evaluated bile salt tolerance.

### Bsh activity

Bsh activity of *E. hirae* F2 was assessed by direct plate Bsh assay method (*Kumar et al., 2010*). 0.1 ml of overnight grown culture was streaked on MRS agar plates supplemented with 0.5% biles like sodium deoxycholate (DCA) and sodium glycocholate (GCA)(Hi-Media, Mumbai, India) and 0.04% $CaCl_2$. Plates were then incubated at 37 °C for 24 to 48 h. Bsh activity was confirmed by the presence of precipitated bile salts around colonies with a silver shining.

### Lipolytic activity

Lipid (Fat & oil) catabolising ability of *E. hirae* F2 by its lipase enzyme was carried out on Spirit Blue Agar (SBA) (Hi-Media, Mumbai, India) plates. 6 mm wells were made on the SBA plates by using sterile cork borer. Wells were filled with 100 µl of fresh culture and incubated at 37 °C for 24–48 h. The presence of lipase was visualized as clear blue colour halos around the wells.

### Acid tolerance

Survival percentage of *E. hirae* F2 under highly acidic environment was assessed by the described method with certain modifications (*Erkkila & Petaja, 2000*). Overnight grown culture of *E. hirae* F2 was inoculated in to fresh MRS broth and grown up to 0.6 to 0.7 OD at 600 nm. The cell culture was centrifuged at 8000 rpm for 10 min at 4 °C (5430R; Eppendorf, Hamburg, Germany). Cell pellet was then washed and resuspended in sterile phosphate buffer saline (pH-7.0). 1 ml of resuspended cells was added into another tube containing 10 ml phosphate buffer saline with different pH: 2, 3, 4 and 7, where, pH 7.0 was served as a control. All the tubes were incubated at 37 °C for 6 h. Calculation of viable cells was carried out by spreading 0.1 ml of culture on MRS agar plates with BCP at 0, 2, 4 and 6 h. All the plates were then incubated at 37 °C for 24 h. Survival percentage of culture

at different pH was calculated by using following formula:

$$\text{Survival \%} = \frac{\text{Final (CFU/ml)}}{\text{Control (CFU/ml)}} \times 100.$$

### Survival and transit tolerance under simulated gastrointestinal tract conditions

Artificial gastrointestinal tract conditions were made up according to the prescribed and published protocol with modifications (*Huang & Adams, 2004*). Gastric juice was made up by mixing of 3 mg of pepsin (Sigma, Bangalore, India), 0.5 gm bile salt (Sigma, Bangalore, India), 1 ml of 0.7% of sodium chloride (Sigma, Bangalore, India) with pH 2–3. Overnight grown culture of *E. hirae F2* was inoculated in to fresh MRS broth and grown up to 0.6–0.7 OD at 600 nm. The cell culture was centrifuged at 8,000 rpm for 10 min at 4 °C. Cell pellet was washed and resuspended in sterile phosphate buffer saline (pH-7.0). 0.5 ml of resuspended cells were added into 2.0 ml of gastric juice and incubated at 37 °C for 4 h. Survival cell count were calculated by spreading 0.1 ml of culture on MRS agar plates at 0, 2, 4 and 6 h. All the plates were incubated at 37 °C for 24 h. Survival percentage under gastric juice was calculated by using following formula:

$$\text{Survival \%} = \frac{\text{Final (CFU/ml)}}{\text{Control (CFU/ml)}} \times 100$$

### Potentiality of E. hirae F2 as yoghurt culture

Curdling ability of *E. hirae* F2 from milk was carried out by using three different categories of milk available in the market (skimmed, semi-skimmed and whole). All the tubes were filled with 10 ml of milk and autoclaved. Autoclaved tubes filled with milk were inoculated with 1 ml of overnight grown culture of *E. hirae* F2 and incubated at 37 °C for 24–48 h.

### Homo/heterofermentative characterization of E. hirae F2

$CO_2$ production from glucose test was applied, in order to determine the homo/heterofermentative characterization of *E. hirae* F2. Citrate lacking MRS broth and inverted Durham tubes were prepared and inoculated with 1% overnight fresh culture. Test tubes were then incubated at 37 °C for 3 days. Gas occurrence in Durham tubes was observed during 3 days, which is the evidence for $CO_2$ production from glucose. This test was further checked out according to the method described by *McDonald et al. (1987)* on homofermentative-heterofermentative differential (HHD) medium. *E. hirae* F2 was streaked on HHD plates and incubate at 37 °C for 24–72 h. In the agar medium, homofermentative colonies were blue to green, while heterofermentative colonies remained white (Fig. 1E).

### In vitro evaluation of antibacterial activity by E. hirae F2

Antibacterial activity of *E. hirae* F2 isolate was tested against different indicator organisms like *Escherichia coli* (MTCC 40), *Staphylococcus aureus* (MTCC 3160), *Salmonella typhi* (MTCC 3215) and *Pseudomonas aeruginosa* (MTCC 424) by using agar well diffusion method on Muller Hinton Agar (MHA) (Hi-Media, Mumbai, India). Overnight grown

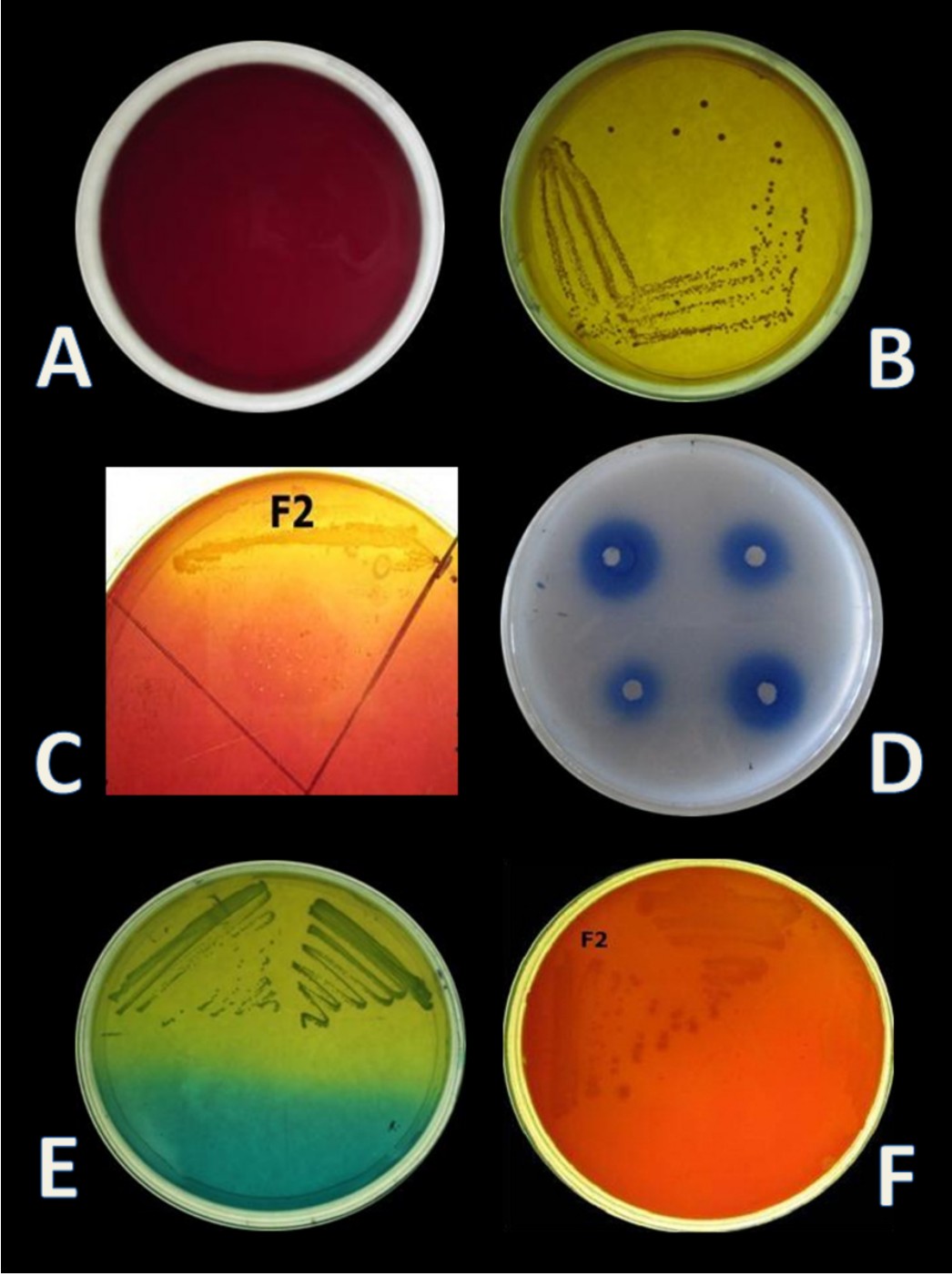

**Figure 1** (A) Normal MRS + BCP agar plate (B) Growth of lactic acid bacteria on MRS + BCP plate (C) Bile salt hydrolase activity of *E. hirae* F2 on MRS-DCA-GCA agar plate (D) Zone of lipid hydrolysis on spirit blue agar plate (E) HHD medium plate showing homofermentative *E. hirae* F2 with green colour colony (F) Gamma-hemolytic activity by *E. hirae* F2 on blood agar plate.

culture of *E. hirae* F2 was centrifuged at 8,000 rpm for 10 min at 4 °C. Supernatant was collected and adjusted to pH 6.7–7.0 and kept at 90 °C for 2 min in water bath. Neutralized supernatant was filter sterilized by 0.2 μm size syringe filter. 500 μl of overnight grown culture of indicator organisms in nutrient broth (Hi-Media, Mumbai, India) were spreaded evenly on to the MHA plates. Then, with the help of a sterile cork borer, wells of 5–6 mm diameter were made in to the plates. A total of 100 μl of supernatants were filled into each of the test wells. A total of 100 μg/ml chloramphenicol antibiotic solution was used as a positive control and sterile MRS broth was used as a negative control. All the plates were incubated at 37 °C for 24 h. Diameter of the zone of inhibition was measured.

### Antioxidative activity against α-Diphenyl- α-Picrylhydrazyl (DPPH)

Antioxidant activity of *E. hirae* F2 against DPPH was carried out according to the method described (*Wu & Pan, 2004*) with certain modifications. The same supernatant, which was used in antibacterial activity, was also used for antioxidant activity. Overnight grown cells were harvested by centrifugation at 6,000 rpm for 10 min at 4 °C. Cell pellet was washed three times and resuspended in phosphate buffer saline (pH-7.0). 100 μl of supernatant and intact cells were separately mixed with 2ml of $6 \times 10^{-5}$M ethanolic solution of DPPH (Sigma, Bangalore, India) and incubated for 30 min in the dark. After 30 min, reduction of DPPH free radicals was measured at 517 nm. Ethanol was used as a blank and DPPH solution without any sample or standard was used as a control. Activity was carried out in triplicates. As a positive control L-ascorbic acid (1 mg/ml) (Sigma-Aldrich, Bangalore, India) was used (*Ahn et al., 2004*). Percentage inhibition of DPPH radical was calculated by using following formula:

$$\% \text{ Inhibition} = \frac{(\text{Absorbance of control} - \text{Absorbance of test})}{\text{Absorbance of control}} \times 100$$

### Antibiotic susceptibility test

Antibiotic sensitivity test was determined by using agar diffusion method of National Committee for Clinical Laboratory Standards (*NCCLS, 1993*). 100 μl of overnight grown culture of *E. hirae* F2 was spreaded on MRS supplemented with BCP agar plate and antibiotic discs were placed on the plate by using sterile forceps. Plates were incubated at 37 °C for 24 h. *E. hirae* F2 was checked for sensitivity or resistance against few antibiotics like, Ampicillin (10 μg), Gentamycin (30 μg), Cephalexin (30 μg), Streptomycin (10 μg), Chloramphenicol (30 μg), Co-Trimoxazole (25 μg) and Tetracycline (25 μg).

### Determination of cell surface hydrophobicity

Cell surface hydrophobicity is the ability of bacterial cell to adhere hydrocarbon. *E. hirae* F2 adhesion to hydrocarbon was determined (*Vinderola & Reinheimer, 2003*). Overnight grown culture was centrifuged at 6,000 rpm for 10 min at 4 °C. Cell pellet was washed and resuspended in phosphate buffer saline (pH 7.0). Absorbance of cell pellet was adjusted to 0.6 OD at 600 nm. 2.0 ml of bacterial suspension with 1.0 ml of xylene (Sigma-Aldrich, Bangalore, India) and 1.0 ml of n-hexadecane (Sigma-Aldrich, Bangalore, India) was mixed altogether by vortexing and then incubated at 37 °C for 1 h. At the end of incubation, two

phases were separated. Aqueous phase was taken out and absorbance was measured at 600 nm. Percentage cell hydrophobicity was calculated by using following formula:

$$\text{Hydrophobicity (\%)} = \frac{[\text{OD (initial)} - \text{OD (final)}]}{[\text{OD (initial)}]} \times 100.$$

## Cell aggregation assay

Auto aggregation assays were performed with some modifications (*Kos et al., 2003*). Freshly grown culture of F2 was centrifuged at 6,000 rpm for 10 min at 4 °C and supernatant was collected in another tube. Cell pellet was washed and resuspended in phosphate buffer saline (pH-7.0). Absorbance was adjusted to 0.6 OD at 600 nm. Then, the dissolved cell pellet was again centrifuged and cell pellet was dissolved in equal amount of previously collected supernatant. It was mixed properly by vortexing and incubated at 37 °C for 3 h. After incubation, 2 ml of the upper layer was taken and absorbance was measured at 600 nm by using broth as a blank (UV-1800, Shimadzu Spectrophotometer, Japan). Cell aggregation percentage was determined with following equation:

$$\text{Formula Aggregation} = \frac{\text{Abs(initial)} - \text{Abs(final)}}{\text{Abs(initial)}} \times 100$$

### *Prevalence of haemolytic and gelatinase activity*

Pathogenicity of *E. hirae* F2 was checked by haemolysis and gelatinase activity. Haemolysis activity of F2 culture was investigated on blood agar (Hi-Media, Mumbai, India) plate supplemented with 5% human blood. Overnight grown culture was streaked on the blood agar plates and incubated at 37 °C for 48 h. After incubation, plates were observed for $\alpha$-haemolysis (partial hydrolysis and greenish zone), $\beta$-haemolysis (clear zone around colony) or $\gamma$-haemolysis (No reaction).

Gelatinase activity was carried out on the nutrient gelatin agar plate. Fresh culture was streaked on to the nutrient gelatin agar plate and incubated at 37 °C for 48 h. After incubation, plates were flooded with the solution of saturated ammonium sulphate. A clear zone around the colonies confirmed presence of gelatinase.

### *Statistical analysis*

All the experiments were carried out in triplicate. The results are presented as mean values and error bars represent standard deviations (SD) of the mean values of results from three replicate experiments.

## RESULTS

### Isolation and screening of potential probiotic lactic acid bacteria from fish gut

Four isolates, designated as F1–F4 were recovered from the gut samples of freshwater fish *Catla catla* on MRS + BCP (bromocresol purple) agar plates. BCP is a pH indicator, which turns in to yellow colour (below pH 5.2) in the presence of lactic acid from its initial

purple colour (above pH 6.8). According to this principle, yellow colour colony of lactic acid bacteria are primarily selected from the MRS + BCP plates (*Badis et al., 2004*) (Figs. 1A, 1B). Out of all isolates, F2 isolate was found to be Gram-positive cocci and catalase negative. F2 strain was then stored for further and long-term study in 20% glycerol at −20 °C and chosen for further probiotic activities and identification.

## Identification of *E. hirae F2* strain by classical and molecular methods

Isolate F2 was identified up to level genus *Enterococcus* according to various biochemical tests (Fig. 2). Isolate F2 was Gram-positive cocci in shape, catalase negative, able to grow at 45 °C and 6.5% NaCl concentration, produced ammonia from arginine. However, the sugar fermentation profile and other various biochemical tests results are represented in Table 1. Isolate F2 was further identified up to species level via 16S rRNA sequencing method by using a pair of universal primer 27f and 1492r. 820 bp sequence of amplified PCR products was obtained after the cycle sequencing (Fig. 3). A total of 820 bp of identified sequence searched for its homology sequence using BLAST in NCBI. 16S rRNA gene sequence of F2 had 99% similarity with *Enterococcus hirae* ATCC 9790 and the sequence was then submitted by our group in the NCBI GenBank database under GenBank ID KF496213.1.

## Probiotic activities of *E. hirae F2*
### Bile salt tolerance

As a potential and possible probiotic, bacteria have to survive in the presence of bile salt in gut. Therefore, whenever probiotics is concerned, bile salt tolerance is the most important criterion for bacteria as a probiotics. Estimated concentration of bile salt in gut is between 0.2 and 2% (*Gunn, 2000*). In this study, bile salt tolerance of *E. hirae* F2 was checked from 0.2 to 2% concentration of bile salts. Results of bile salt tolerance are represented in Table 2. Results showed that *E. hirae F2* was able to survive under the concentration of 0.2–1.0% of bile salts (Fig. 4). However, viability of *E. hirae* F2 cells were started to decrease after 120 min under 0.6% of bile salt. Under 0.8% of bile salt concentration, cell viability was started to decrease after 90 min, whereas cell viability was gradually decreased in 1.0% concentration of bile salts after 30, 60, 90 and 120 min.

### Screening of Bsh and lipolytic activity

*E. hirae F2* was also screened for the presence of Bsh enzyme. Growth of *E. hirae* F2 on MRS agar plate supplemented with DCA and GCA and the presence of precipitated bile salts around the colony with a shine confirmed that, *E. hirae* F2 was able to hydrolyse DCA and GCA via the production of Bsh enzyme (Fig. 1C). As a part of normal human metabolism, lipase is an essential enzyme which hydrolyse fat into small components that are readily absorbed by the intestines (*Somboon, Mukkharin & Suwimon, 2015*). The ability of *E. hirae* F2 to produce lipolytic enzyme was examined on spirit blue agar medium containing emulsified lipid with the spirit blue dye. Lipolytic activity of *E. hirae* F2 is shown with blue colour halos around the well indicating lipolysis i.e., metabolizing the lipid in the medium (Fig. 1D).

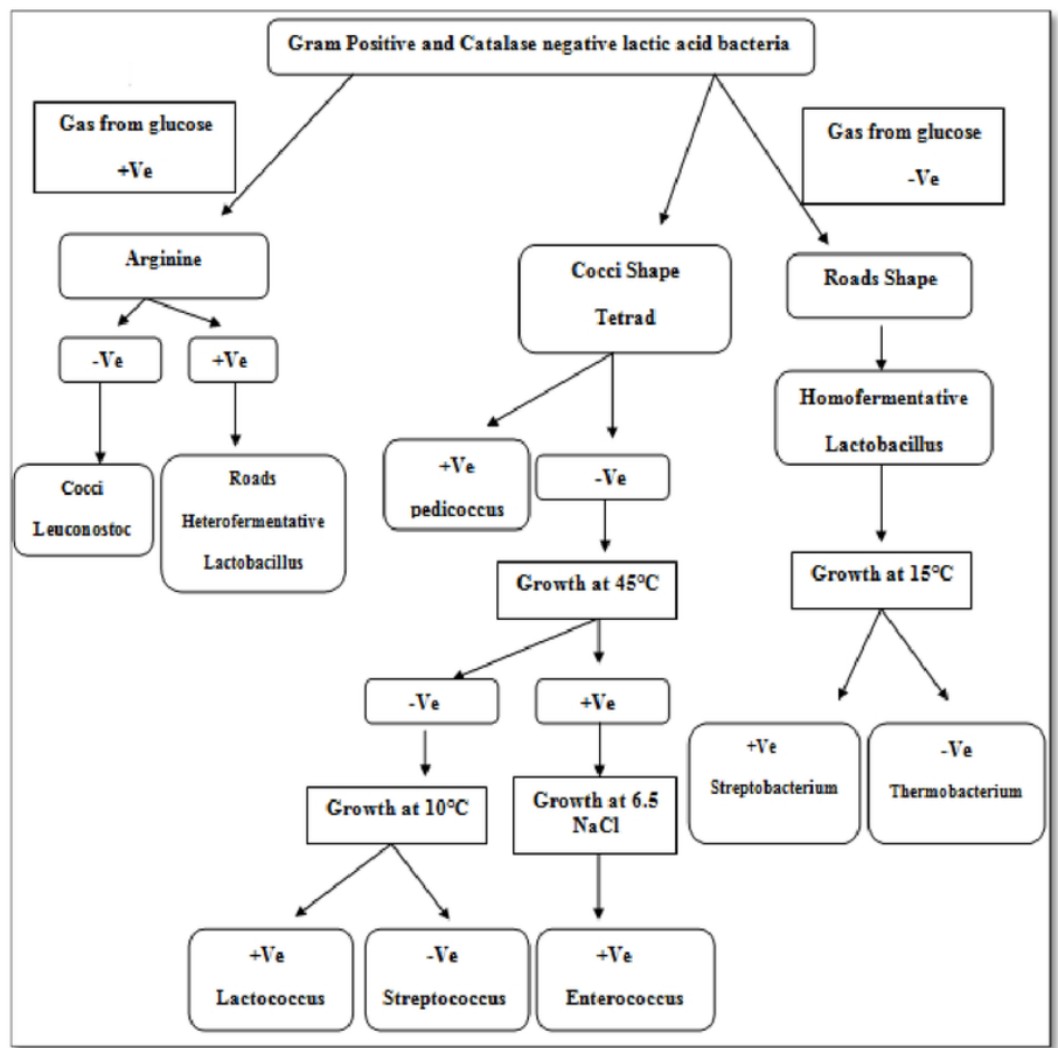

**Figure 2** Route for identification of lactic acid bacteria at genus level (*Salminen et al., 2002*).

### Acid tolerance response and survival under artificial gastrointestinal tract conditions

Due to the high acidic condition in gut, bacteria to work as a probiotics must have to survive under such kind of conditions. Therefore, after confirmation of bile salt tolerance, *E. hirae* F2 was further tested to prove its ability to survive/tolerate under highly acidic environment. *E. hirae* F2 was tested at pH 2.0, 3.0, 4.0 and 7.0 as a control. Number of *E. hirae* F2 cells survived after 2, 4, and 6 h of exposure at different pH of 2, 3 and 4 are presented in Fig. 5. *E. hirae* F2 had no viability after 6 h but shown lower viability after 2 and 4 h at pH 2. A good viability with gradually decreased number of colonies was recorded after 2, 4 and 6 h at pH 3. Whereas, good growth and viability was seen even after 6 h of incubation at pH 4. *E. hirae* F2 had highest viability at pH 7 and lowest viability at pH 2. Endurance of *E. hirae* F2 under artificial gastrointestinal tract conditions was carried out according to (*Huang & Adams, 2004*) with modifications with pH between 2 and 3 at

**Table 1    Biochemical characterization of F2 isolate.**

| Characteristics | | Isolate F2 |
|---|---|---|
| Colony appearance | | Circular, smooth, entire |
| Gram's nature | | Gram positive cocci |
| Catalase test | | − |
| Growth at 45 °C | | + |
| Growth at 6.5% NaCl concentration | | + |
| Ammonia production | | + |
| **Sugar fermentation profile** | | |
| Glucose | Gas | − |
| | Acid | + |
| Fructose | Gas | − |
| | Acid | + |
| Sucrose | Gas | − |
| | Acid | + |
| Maltose | Gas | − |
| | Acid | + |
| Lactose | Gas | − |
| | Acid | + |
| Mannose | Gas | − |
| | Acid | + |
| Mannitol | Gas | − |
| | Acid | + |
| Ribose | Gas | − |
| | Acid | + |
| Na gluconate | Gas | − |
| | Acid | − |
| Xylose | Gas | − |
| | Acid | − |
| Inositol | Gas | − |
| | Acid | − |

37 °C for 6 h and is presented in Fig. 6. *E. hirae* F2 survived under such type of conditions up to 2 h without a higher loss in cell viability. Whereas, viability started decreasing after 4 and 6 h. This result indicates that *E. hirae* F2 is able to survive efficiently up to 2 h under gastrointestinal tract conditions without loss in cell viability.

### Potentiality test of E. hirae F2 as starter culture for yoghurt formation and identification of homo/heterofermentation

Curd formation ability of *E. hirae* F2 is represented in Table 3. It showed excellent curd formation after 72 h in all three kinds of milk. This test indicates that *E. hirae* F2 might be used as a starter culture for the production of probiotic yoghurts or other dairy products. Two tests were performed for the identification of homo/heterofermentation, production of $CO_2$ from glucose and streaked on fructose + bromocresol green plates. On

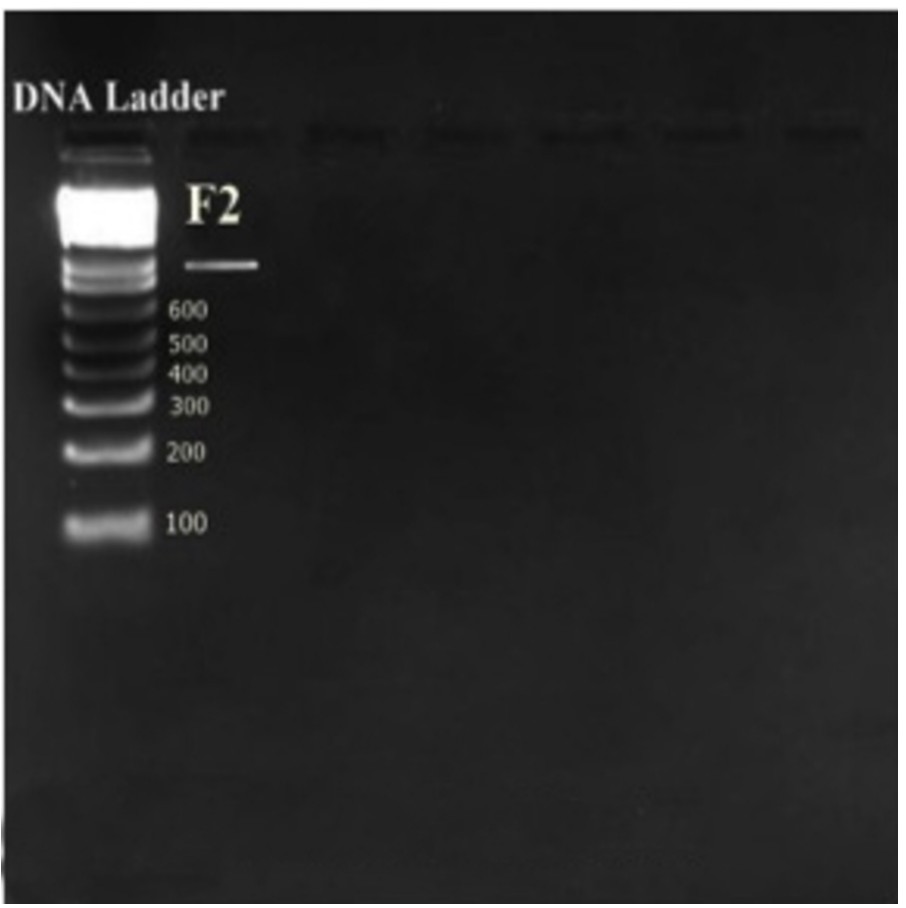

**Figure 3** 16S rDNA fragment of *E. hirae* F2 amplified by PCR.

**Table 2** Viability result of bile salt tolerance.

| Bile salt concentration (%) | Result | Viability (%) | | | |
|---|---|---|---|---|---|
| | | 30 min | 60 min | 90 min | 120 min |
| 0.2 | + | 100% | 100% | 100% | 100% |
| 0.4 | + | 100% | 100% | 100% | 100% |
| 0.6 | + | 100% | 100% | 100% | 88.7% |
| 0.8 | + | 100% | 100% | 82.6% | 62% |
| 1.0 | + | 80% | 67.4% | 45% | 20.6% |
| 1.2 | − | No growth | No growth | No growth | No growth |
| 1.4 | − | No growth | No growth | No growth | No growth |
| 1.6 | − | No growth | No growth | No growth | No growth |
| 1.8 | − | No growth | No growth | No growth | No growth |
| 2.0 | − | No growth | No growth | No growth | No growth |

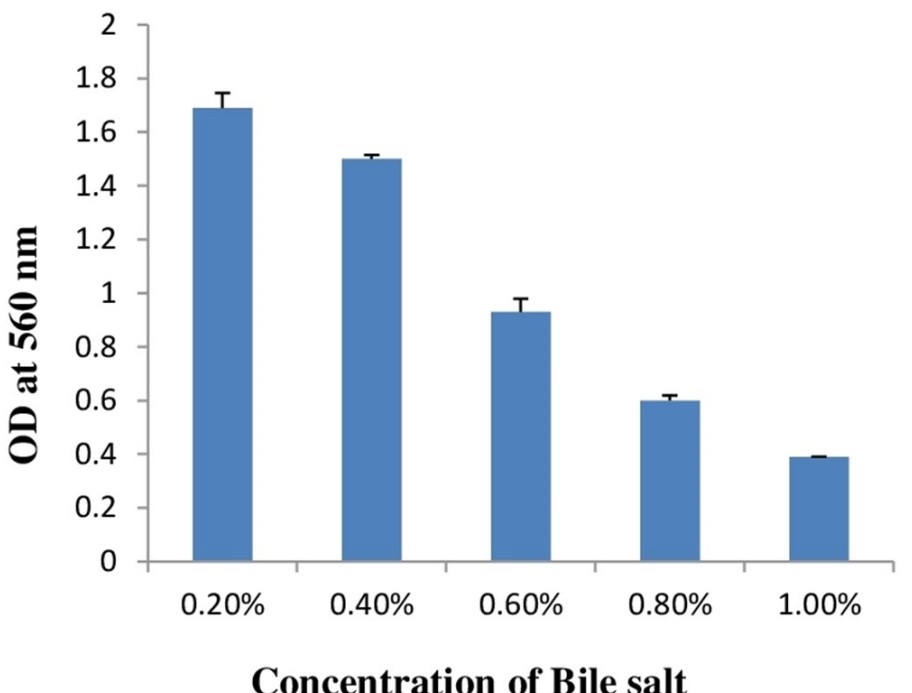

**Figure 4 Bar graph showing bile salt tolerance of *E. hirae* F2 in MRS broth.** Error bars represent SD of the mean values of results from three replicate experiments.

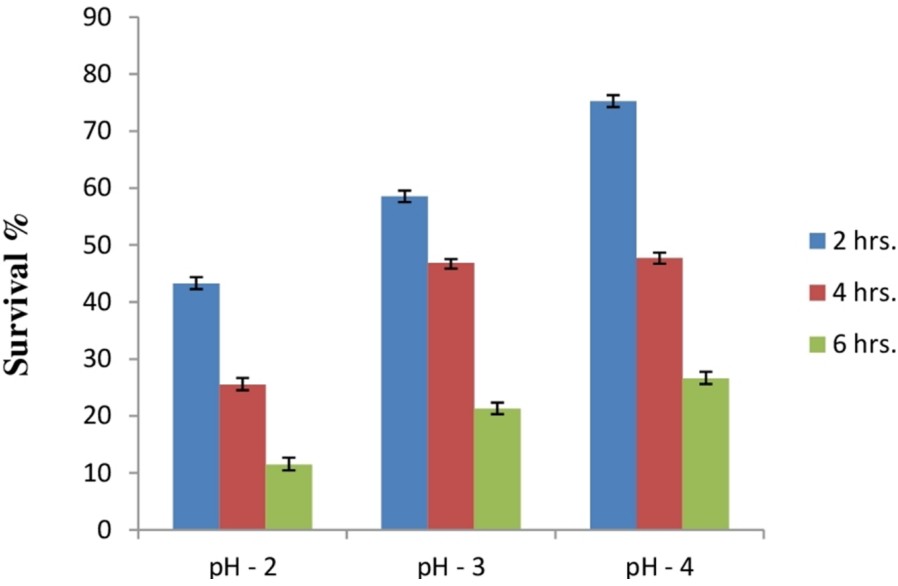

**Figure 5 Bar graph showing acid tolerance response of *E. hirae* F2 at different pH.** Error bars represent SD of the mean values of results from three replicate experiments.

the basis of this test *E. hirae* F2 was considered as a homofermentative. The results of the homo/heterofermentation behaviour of *E. hirae* F2 are represented in Table 4.

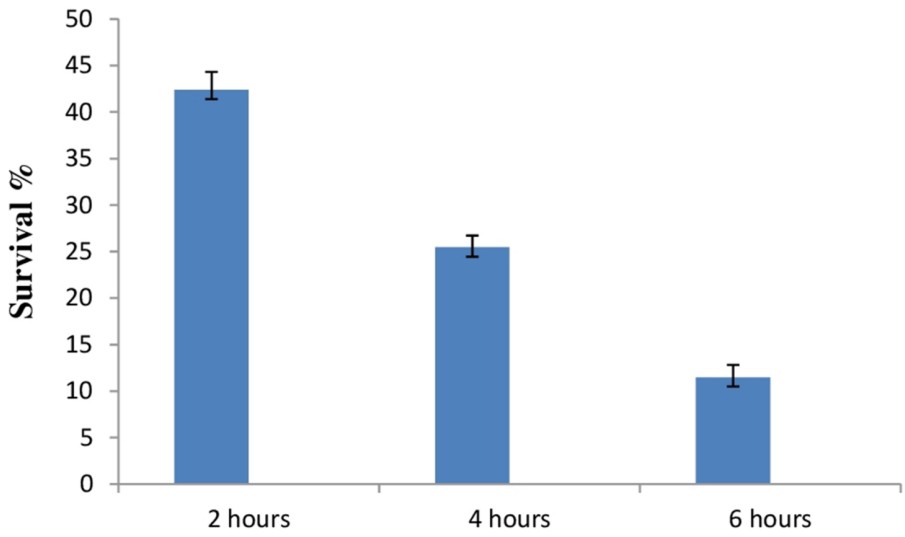

**Figure 6** **Bar graph showing ability of *E. hirae* F2 to survive under artificial gastrointestinal tract conditions.** Error bars represent SD of the mean values of results from three replicate experiments.

**Table 3** **Result of curd formation.**

| Isolate | Skimmed milk | | | Semi-skimmed milk | | | Whole milk | | |
|---------|------|------|------|------|------|------|------|------|------|
| | 24 h | 48 h | 72 h | 24 h | 48 h | 72 h | 24 h | 48 h | 72 h |
| F2 | + | ++ | +++ | − | + | +++ | ++ | ++ | +++ |

Notes.
+++, Excellent curd formation; ++, Very good curd formation; +, Good curd formation; −, No curd formation.

**Table 4** **Result of homo/heterofermentation with phenotypic description.**

| Isolate | $CO_2$ from Glucose | Phenotypic description on Plate |
|---------|---------------------|--------------------------------|
| F2 | No gas, growth present | Green colour colony with yellow background |

## Antibacterial activity

Lactic acid bacteria can inhibit growth of diverse pathogenic bacteria with different mechanisms. Therefore, *E. hirae* F2 was studied for its antagonistic effect against several pathogenic bacteria like *E. coli, S. aureus, S. typhi and Pseudomonas spp.* Results of the antibacterial activity of *E. hirae* F2 are represented in Fig. 7 in the form of zone of inhibition. *E. hirae* F2 had significant antagonistic effect against all tested strains.

## Antioxidative activity against DPPH

DPPH is generally used as a substrate to assess the antioxidant activity of medicinally important constituents. It is also a well-known free radical and a scavenger. DPPH has a deep purple colour in ethanolic solution and has a strong absorption at 517 nm (*Sharma & Bhat, 2009*). Principle of this assay is based on the changing of DPPH radical colour from purple to yellow in the presence of antioxidant component. Whenever, antioxidant molecule reacts with DPPH radicals, antioxidant molecule donates hydrogen atoms in solution, which leads to the formation of non-radical form of DPPH-H. This formation

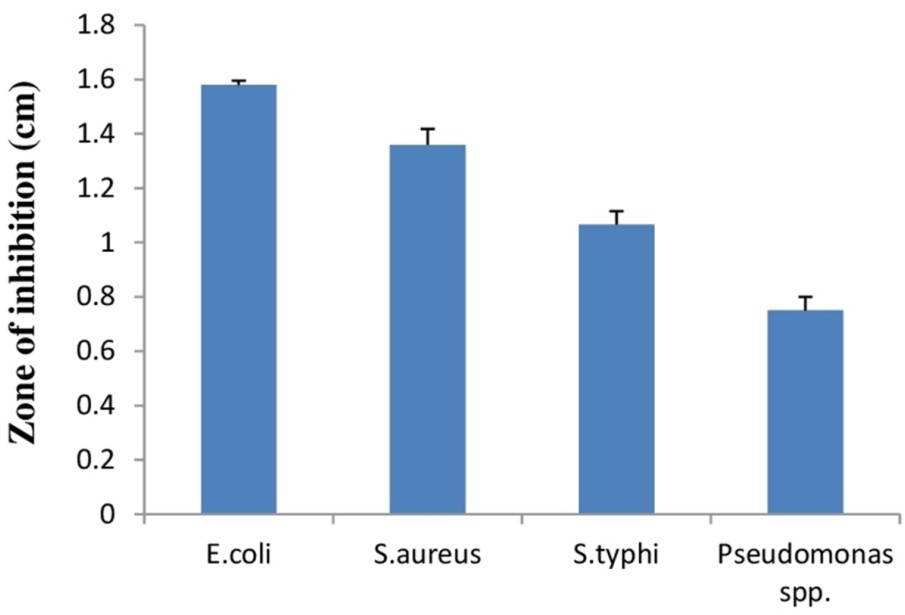

**Figure 7** **Bar graph showing antibacterial activity of *E. hirae* F2 against *E. coli*, *S. aureus*, *S. typhi*, and *Pseudomonas spp*.** Error bars represent standard SD of the mean values of results from three replicate experiments.

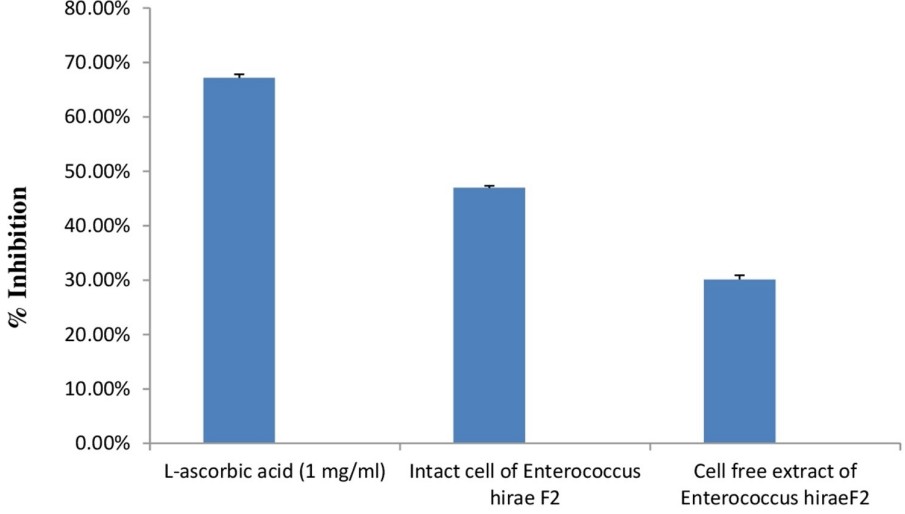

**Figure 8** **Bar graph showing antioxidant activity of *E. hirae* F2 against DPPH.** Error bars represent SD of the mean values of results from three replicate experiments.

results in the scavenging of the radicals from 2,2-Diphenyl-1-picrylhydrazyl (purple colour) to diphenylpicrylhydrazine (yellow colour) (*Mohammad et al., 2009*). Results of the antioxidant activity of intact cells and cell free extract of *E. hirae* F2 are represented in the Fig. 8. However, antioxidant activity of intact cells of *E. hirae* F2 was higher than the supernatants (cell free extracts). Scavenging ability of intact cells against DPPH radical was 47.43% compared to 30.45% for cell free extracts (Fig. 8).

**Table 5  Result of antibiotic susceptibility test.**

| Antibiotic | Concentration ($\mu$g) | *E. hirae* F2 |
|---|---|---|
| Ampicillin | 10 | R |
| Gentamycin | 10 | S |
| Cephalexin | 30 | R |
| Streptomycin | 10 | S |
| Chloramphenicol | 30 | S |
| Co-Trimoxazole | 25 | S |
| Tetracycline | 25 | S |

**Notes.**

R, Resistance; S, Sensitive.

### Antibiotic susceptibility test

Usually the antibiotic susceptibility test is carried out to evaluate the potentiality of particular antibiotic for the treatment of bacterial infection successfully. In the context of probiotics, antibiotic susceptibility test is carried out to find sensitivity and resistance of probiotic strain against particular antibiotics taken in to the gut. The antibiotic susceptibility profile of *E. hirae* F2 is shown in Table 5. The profile of the antibiotic susceptibility of *E. hirae* F2 indicates that strain was resistant to Ampicillin and Cephalexin. Sensitive to Gentamycin, Streptomycin, Chloramphenicol, Co-Trimoxazole and Tetracycline. Resistance of *E. hirae* F2 against specific antibiotic indicates that, it can be used as a probiotic at the same time when that antibiotic is present in to the gut or treatment with those antibiotics is required.

### Cell surface hydrophobicity and cell aggregation assay

Cell surface hydrophobicity and cell aggregation are two important phenotypic characteristics of a potential probiotic strain (*Tamang et al., 2009*). In bacterial cells, variety of mechanisms plays an important role in adhesion of bacterial cells to the intestinal cells. Out of this, bacterial cell surface hydrophobicity and cell aggregation are most important. Bacterial attachment to the host cells is due to the hydrophobic surface of bacteria and the activity of aggregation. Cell surface hydrophobicity and aggregation are considered as an important factor for the maintenance of probiotic bacteria in the gastrointestinal tract of humans (*Kiely & Olson, 2000*). Figure 9 shows percent cell surface hydrophobicity of *E. hirae* F2 against xylene, n-hexadecane and cell auto aggregation. Cell surface hydrophobicity of *E. hirae* F2 in the presence of xylene was 38.7%, which is higher than the n-hexadecane i.e., 34.4%. Cell auto aggregation of *E. hirae* F2 was calculated 28.4%.

### Prevalence of haemolytic and gelatinase activity

*E. hirae* F2 had no positive activity for haemolysis as well as for gelatinase activity which is considered as gamma-haemolysis on blood agar plate and non-pathogenic (Fig. 1F).

## DISCUSSION

In the present study, we isolated and characterize *E. hirae* F2 from the gut of freshwater fish, *Catla catla*, which showed desirable characteristics to be a potent probiotic strain.

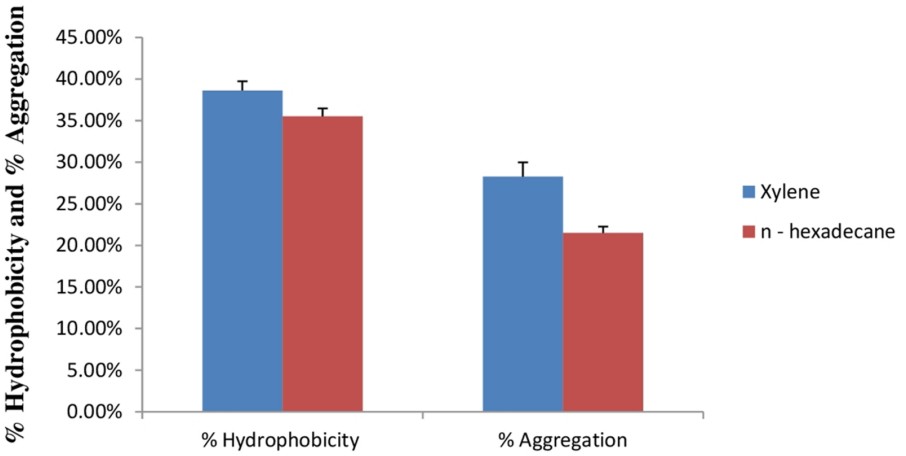

**Figure 9** **Bar graph showing relationship between hydrophobicity (%) and auto aggregation ability (%).** Error bars represent SD of the mean values of results from three replicate experiments.

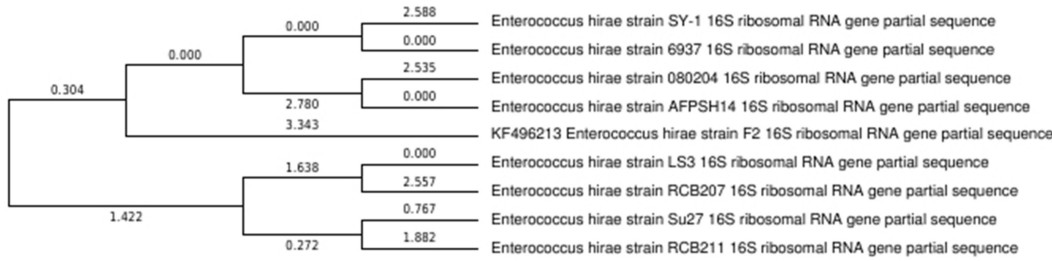

**Figure 10** **Phylogenetic tree showing relationship between *E. hirae* F2 and other *Enterococcus hirae* strains.**

Despite the fact that there have been studies on the utilization of bacteria from this genus as probiotics, there are no reports of its isolation from the digestive tract of fish. This is the first attempt to illustrate the potentiality of *E. hirae* as a probiotics isolated from the gut of fish with complete experimental evidence, which was not carried out in the previous studies. High prevalence of *E. hirae* F2 in the gut of fish might be attributed to their invulnerability to acidic and bile salt conditions. Isolated strain was Gram's positive cocci, catalase negative, and was able to grow up to 45 °C and under 6.5% NaCl concentration. It also produces ammonia from arginine, which is an important activity of lactic acid bacteria and can carry out various carbohydrate fermentation. Results obtained from the molecular analysis confirmed the bacterial strains isolated in this work as *E. hirae*. The phylogenetic tree shows that the *E. hirae* F2 was very closely related to others *E. hirae* strains (Fig. 10). Resistance to pH and bile salts is an important factor to select isolates with probiotic characteristics. Bile salts are a segment of bile, which is secreted from the gallbladder on the assimilation of cholesterol and entering into the small intestine (*Rushdy & Gomaa, 2013*). Results of this study validated that *E. hirae* F2 was surviving bile salts from 0.2% to 1.0% concentration, which is 4–5 times higher than the bile salt concentration of human intestine. However, previous studies (*Gupta & Tiwari, 2015*; *Selvaraj et al., 2014*) on *E.*

*hirae* also indicated similar kind of results of bile salt tolerance. Further confirmation related to bile salt tolerance of *E. hirae* F2 was confirmed by the presence of BSH enzyme. BSH enzyme has ability to hydrolyse bile salts and activity of BSH enzyme in *E. hirae* was not carried out by any of the previous studies. This study signifies the presence of BSH enzyme activity first time. *E. hirae* F2 was able to hydrolyse bile's like sodium deoxycholate and sodium glycocholate. Besides the tolerance of bile salts and the presence of BSH activity, *E. hirae* F2 also survive high acidic condition of the gut. It was found to be viable up to 4 h in pH 2, and remain viable even after 6 h under pH 4. Out of this, when exposed to artificial gastrointestinal tract conditions which contains bile salts, pepsin and NaCl with pH in between 2 and 3, *E. hirae* F2 exhibited a survival rate of 44.41% after 2 h. This activity indicating that *E. hirae* F2 can be classified as tolerant to the gastrointestinal secretions. In general, the acid tolerance depends upon the pH profile of Hþ-ATPase and the association of the cytoplasmic membrane. This is influenced by the type of bacterium, organization of development medium and the incubation conditions (*Madureira et al., 2011*). Selection of potential probiotic strains influenced with the presence of high concentration of bile components and acidity in the proximal intestine of the host (*Hyronimus, Le & Urdaci, 1998*).

However, when we administer any bacteria orally, it must face acidic pH (2–3) of gastric juice, high bile salt concentration, pepsin and so on. These conditions normally inhibit the growth of ingested microorganisms. Therefore, probiotic bacteria need to survive under these conditions. Our study showed that, if *E. hirae* F2 strain was taken orally as a probiotics, than it will survive under human gastrointestinal conditions when it passes through the stomach and would attain its probiotic activities. This strain might be used as a future probiotics for the prosperity of human health. According to the guidelines of FAO/WHO, determination of haemolytic action is considered as a safety perspective for the choice of probiotic strains. This activity was also investigated and explored in this study. Results demonstrated the absence of haemolytic activity.

To be a beneficial probiotic, the antimicrobial and antioxidant activity is also a vital criterion in selecting the desired strain. *E. hirae* F2 was able to inhibit the growth of pathogens like, *E. coli, S. aureus, S. typhi* and *Pseudomonas spp.* In this study, antibacterial activity of *E. hirae* F2 might be considered due to the Hiracin S, due to the production of organic acids, or by $H_2O_2$ production. However, *E. hirae* F2 also produces lactic acid, which can serve as an antibacterial agent. This ability of *E. hirae* F2 might help in maintaining the healthy normal flora of human gut. Antioxidant activity by *E. hirae* F2 was never assessed by the previous studies. Antioxidant molecules helps the body in neutralizing and removing the free radicals from the body and prevent from various diseases and conditions like cancer (*Woodmansey, 2007*), diabetes (*Lim et al., 2013*), as well as lowering the process of ageing (*Yadav et al., 2009*) . Moreover, antioxidants help in immune system support, skin benefits, eye problems and in mood disorders. *E. hirae* F2, cell free extract as well as intact cell culture shows good antioxidant activity against DPPH. However, intact cell culture showed better activity than cell free extract. Apart from antibacterial and antioxidant activity, antibiotic susceptibility test was performed. *E. hirae* F2 was found to be sensitive to Gentamycin, Streptomycin, Chloramphenicol, Co-Trimoxazole and Tetracycline and

resistant to Ampicillin and Cephalexin. This indicates the use of straind when a particular antibiotic is present in the gut or treatment with those antibiotics is required.

Adherence to the mucosal surface as well as to epithelial cells in intestine is also considered as one of the most significant quality of probiotic bacteria. Probiotic bacterial attachment on to the epithelial surface of the gut is important for their colonization and for providing their probiotics attributes in to the gut of host (*Pedersen & Tannock, 1989*; *Freter, 1992*). Aggregation ability of a probiotic bacterium also plays a vital role in the elimination of pathogens into the gut by forming biofilms in to the host (*Austin et al., 1998*). However, a probiotic bacterium with cell surface hydrophobicity and aggregation ability is more competent to adhere in to the gut. *E. hirae* F2 showed a significant relationship between cell surface hydrophobicity and aggregation. This relationship of *E. hirae* F2 indicates that cell surface hydrophobicity and aggregation ability is helpful in creating a microenvironment in to the gut by adhering to the epithelial surface and excrete its antibacterial substances like Hiracin S or lactic acid which may inhibit the growth of pathogenic bacteria in the gut.

Lipase is an important enzyme for the digestion of lipid substances in body. Lipase carries out hydrolysis of fat in to small components, which can be easily absorbed through intestine. In the absence of lipase, excess amount of fat in to the body is responsible for the heart diseases, diabetes and various other health effecting diseases and conditions. Therefore, lipase is needed to lower down the risk of such kind of diseases. Furthermore, lipase also helps in preventing, from excess weight gain, obesity, maintaining pancreatic enzymes at optimum level, boost immune function, boost absorption of vitamins and minerals etc. (*Edward, 2011*). This study reveals that *E. hirae* F2 had a good lipase activity and able to hydrolyse lipid. Therefore, if *E. hirae* F2 taken as probiotic, it will produce lipase in the gut and will help in maintaining body fat as well as, it will promote other health benefits.

Lactic acid bacteria are mainly categorized as a homo/heterofermentative according to their by-products of sugar fermentation. Lactic acid is produced from glucose as the primary by-product by homofermentative lactic acid bacteria. Whereas, ethanol, lactic acid, acetic acid and carbon dioxide ($CO_2$) is produced as by-products from glucose by heterofermentative lactic acid. On the basis of this, *E. hirae* F2 was identified as a homofermentative. As an homofermentative, the curd formation ability of *E. hirae* F2 also suggested that it will be used as a starter culture for curd formation from the milk. This activity of *E. hirae* F2 can be used in dairy industries as a starter culture, which will grow under controlled conditions to modify the texture and gives a flavor to the final product. These products may be used as probiotics, which impart the health benefits.

## CONCLUSIONS

In summary, the outcomes in this study claim that *E. hirae* F2 is a resistant strain against acidic conditions, bile salts and simulated gastric juice. With cell surface hydrophobicity and aggregation with good curd formation ability and BSH, lipase, antimicrobial, antioxidant activities, *E. hirae* F2 exhibited desirable probiotic properties *in vitro*. All of these probiotic

activities of *E. hirae* F2 indicated that it might be used as a new promising probiotic strain in the future.

### Funding
The authors received no funding for this work.

### Competing Interests
The authors declare there are no competing interests.

### Author Contributions
- Mohd Adnan conceived and designed the experiments, performed the experiments, analyzed the data, contributed reagents/materials/analysis tools, wrote the paper, prepared figures and/or tables, reviewed drafts of the paper.
- Mitesh Patel conceived and designed the experiments, performed the experiments, wrote the paper, prepared figures and/or tables, reviewed drafts of the paper.
- Sibte Hadi analyzed the data, contributed reagents/materials/analysis tools, reviewed drafts of the paper.

### DNA Deposition
The following information was supplied regarding the deposition of DNA sequences:
 GenBank: KF496213.1.

### Data Availability
 The raw data has been supplied as a Supplemental Information 1.

### Supplemental Information
Supplemental information for this article can be found online at http://dx.doi.org/10.7717/peerj.3085#supplemental-information.

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
