# Peer review of "Functional and health promoting inherent attributes of Enterococcus hirae F2 as a novel probiotic isolated from the digestive tract of the freshwater fish Catla catla"

_PeerJ, doi:10.7717/peerj.3085_

## Round 0.1 · original submission · Major Revisions

· Academic Editor

Major Revisions

I have received two different reviews but my opinion is in the middle of the two as I find it has scientific merit. If you can thoroughly and clearly reply, with special attention to the major concerns of the first review, in a point-by-point form along with the revised manuscript, then your paper can be reconsidered for publication.

Reviewer 1 ·

Basic reporting

no comment

Experimental design

no comment

Validity of the findings

no novelty

Additional comments

The revised manuscripts focuses on the isolation and identify novel probiotic Enterococcus strain from the gut of Catla catla fish and evaluates its potentiality as a potent probiotics. The study, despite it has hard work, it is difficult to identify the novelty in the paper. And, these are some important strengths of the study:
1. The probiotics used in humans commonly come from dairy foods, whereas the sources of probiotics used in animals are often the animals’ own digestive tracts. Why the author chose Enterococcus hirae F2 from fresh water fish as a probiotics for humans?
2. Enterococcus is well known as the normal resident in human gastrointestinal tract and successfully used as probiotic preparations in clinical treatment. However, Enterococcus species are recognized as nosocomial pathogens, which have virulence genes and resistance to certain antibiotics.
Enterococcus hirae F2, a strain isolated from digestive tract of fresh water fish Catla catla, which virulence and antibiotic resistant phenotypes (cytolysin and gelatinase production, antibiotic susceptibility) and genes (cylA, gelE, ace, agg, esp, and vanA) should be surveyed and the results must indicate that the tested virulence determinants are nontoxic.
3. Line 117:“Fresh-water fishes Catla catla (1.5 to 2 kg), were collected from the local fish market.” Consider the source, there are two categories of sources of fish: farmed or wild. If the fishes are farmed ones, whether some probiotic products were used during aquaculture?
4. However, in vivo studies are needed to determine the safety of the strain.

Reviewer 2 ·

Basic reporting

ok

Experimental design

ok

Validity of the findings

ok

Additional comments

I reviewed the paper entitled 'Functional and health promoting inherent attributes of
Enterococcus hirae F2 as a novel probiotics isolated from
digestive tract of fresh water fish Catla catla'. The paper reported Enterococcus hirae F2 as novel probiotics which can utilized as dietary supplement. Overall the manuscript is organized nicely and presented the magnitude of data required for publication. However following are some of my suggestions to improve the quality of manuscript:
Major concerns
1) The abstract lacks specific methodology used for the study
2) Materials and methods part is written nicely
3)Quality of figure is a question mark (comments incorporated in paper)
4) Tables should follow the standard three line table format
Minor points:
the minor changes have been incorporated in the manuscript as track changes.

Annotated reviews are not available for download in order to protect the identity of reviewers who chose to remain anonymous.

---

## Round 0.2 · accepted · Accept

· Academic Editor

Accept

Thank you for your revision submission. I have evaluated it and am happy to accept it.